# Exploratory Study on Nanoparticle Co-Delivery of Temozolomide and Ligustilide for Enhanced Brain Tumor Therapy

**DOI:** 10.3390/pharmaceutics17020191

**Published:** 2025-02-04

**Authors:** Gang Ke, Mingxia Zhang, Pengyi Hu, Jing Zhang, Abid Naeem, Lianfang Wang, Huixin Xu, Yu Liu, Ming Cao, Qin Zheng

**Affiliations:** 1Key Laboratory of Modern Preparation of Traditional Chinese Medicine, Ministry of Education, Jiangxi University of Chinese Medicine, Nanchang 330004, China; kegang@jxutcm.edu.cn (G.K.); zhangmingxia@jxutcm.edu.cn (M.Z.); 20070972@jxutcm.edu.cn (P.H.); jing.zhang@jxutcm.edu.cn (J.Z.); wanglianfang1@jxutcm.edu.cn (L.W.); xuhuixin@jxutcm.edu.cn (H.X.); liuyu@jxutcm.edu.cn (Y.L.); caoming@jxutcm.edu.cn (M.C.); 2Luzhou People’s Hospital, Luzhou 646000, China; 3School of Life Science, Advanced Research Institute of Multidisciplinary Science, School of Medical Technology, Key Laboratory of Molecular Medicine and Biotherapy, Key Laboratory of Medical Molecule Science and Pharmaceutics Engineering, Beijing Institute of Technology, Beijing 100081, China; 7520240013@bit.edu.cn

**Keywords:** glioblastoma multiforme, temozolomide, ligustilide, dual drug-loaded nanoparticles

## Abstract

**Background:** Temozolomide (TMZ) is the first-line therapy for glioblastoma (GBM), but its clinical efficacy is limited by its short half-life, poor brain targeting, adverse side effects, and the development of drug resistance. Ligustilide (LIG) has been shown to enhance blood-brain barrier permeability and reduce P-glycoprotein activity, thereby potentiating the synergistic effect of TMZ against GBM. **Methods:** The dual-drug-loaded nanoparticles encapsulating both TMZ and LIG (TMZ/LIG-NPs) were prepared using Poly (d,l-lactic-co-glycolide)-monomethoxy poly (ethylene glycol) (PLGA-mPEG). The physicochemical properties of the NPs, including particle size and zeta potential, were characterized. Cellular uptake of NPs was evaluated using flow cytometry and fluorescence staining. The pharmacokinetic profile and cytotoxicity of TMZ/LIG-NPs were compared to those of free TMZ and a mixture of TMZ and LIG in rat and glioma cells, respectively. **Results:** The mean particle size of TMZ/LIG-NPs was 117.6 ± 0.7 nm, with a zeta potential of −26.5 ± 0.4 mV. Cellular uptake of NPs was significantly higher than that of free drug in U251 cells. Encapsulation of TMZ in NPs significantly increased its half-life by 1.62-fold compared to free TMZ and significantly improved its pharmacokinetic profile. Moreover, the storage stability of the TMZ/LIG-NPs solution was extended to one month. The toxicity of TMZ/LIG-NPs to glioma cells C6 and U251 was markedly enhanced compared to the mixture of TMZ and LIG. **Conclusions:** The development of TMZ/LIG-NPs using PLGA-mPEG effectively enhanced the stability and efficacy of both TMZ and LIG. This dual drug-loaded nanoparticle system represents a promising strategy for glioblastoma therapy.

## 1. Introduction

Glioblastoma (GBM) is the most common and aggressive type of primary brain tumor [1], with patients having a median survival of only 12–15 months [2,3]. It has been classified as a grade-IV tumor by The World Health Organization [4]. The first-line treatment for GBM starts with safe resection, followed by combined radiotherapy and chemotherapy.

Temozolomide (TMZ) (Figure 1A) is one of the few drugs with antitumor activity in the central nervous system [5,6]. TMZ is the first-line chemotherapeutic option after surgery and radiotherapy for GBM [7,8]. It is used most commonly as maintenance therapy [9] and is the most efficacious drug for GBM. Two-year survival after radiotherapy combined with TMZ (26.5%) is two times that of use of radiotherapy alone (10.4%) [10,11,12]. However, with the prolonged use of TMZ, 50% of patients exhibit resistance, resulting in diminished therapeutic effects [13]. Resistance to TMZ has become an obstruction to clinical therapy.

Ideally, chemotherapeutic agents should exhibit a targeted distribution at neoplastic foci, sustain an extended therapeutic window, achieve optimal toxic efficacy against tumor cells, and concurrently mitigate adverse effects on healthy tissues [14]. However, numerous factors affect TMZ treatment for GBM: the existence of the blood-brain barrier (BBB), tumor heterogeneity, and drug resistance. The antitumor effect of TMZ leads to cell death by inducing DNA degradation by its methylation via its hydrolyzate 5-(3-methyl-1-triazen-1-yl) imidazole-4-carboxamide (MTIC). However, the low permeability of the BBB results in the inability of MTIC to penetrate it [15,16], and only the metabolites of TMZ within the brain have antitumor effects. The oral bioavailability of TMZ is 100%, but only 20% of systemic TMZ reaches brain tissue [17]. Moreover, TMZ in the blood would damage hematopoietic stem cells indiscriminately, leading to dose-limiting hematological toxicity in patients [18]. Therefore, increasing TMZ accumulation in the brain can enhance the efficacy and reduce the side-effects of TMZ.

Studies have indicated that combining traditional Chinese medicine formulations (TCMFs) with chemotherapy can enhance efficacy, reduce resistance, improve drug performance, and increase patient tolerance [19]. Within the TCM pharmacopeia, a subset of herbal remedies has been posited to enhance the cerebrovascular permeation of concomitant therapeutic agents, a pharmacological action referred to as “cerebro-directing”, which guide the upward movement of drugs can improve drug permeability across the BBB and tumor barriers [20,21]. Notable examples include Acorus tatarinowii, Dryobalanops aromatica, and Ligusticum chuanxiong, which have been utilized historically for their purported cognitive and cerebrovascular benefits [22,23]. Some guiding meridian Chinese medicines can be applied to tumor therapeutic formulations to enhance the uptake and targeting of antitumor drugs [24]. This may potentially increase the antitumor efficacy of drugs and reduce the tumor’s multidrug resistance.

*Ligusticum chuanxiong* Hort. has the effects of “activating blood circulation to remove blood stasis and guiding the medicine upward”. Ligustilide (LIG; Figure 1B) is one of its active ingredients. Previously, our findings indicated that combining TMZ with LIG could enhance the anti-GBM effect of TMZ, and LIG can enhance the anti-glioma effects of TMZ through the PI3K/Akt signaling pathway, indicating the potential of LIG to overcome resistance in GBM [25]. Moreover, we demonstrated that LIG can inhibit the P-glycoprotein (P-gp) drug-efflux system and tight-junction proteins zonula occludens-1 (ZO-1) and claudin-5, thereby further increasing the permeability of the BBB and the transmembrane transport of pharmaceutical agents [23,26]. Studies have indicated that LIG inhibits tumor growth by targeting angiogenesis in tumor tissues [27]. Research on the combined use of TCMF ingredients with multi-target effects that enhance BBB permeability and synergistically combat tumors with TMZ is limited. To exert the synergistic effect of the two drugs, an effective co-delivery strategy to the brain is needed.

“Nanomedicine” offers the advantages of superior biocompatibility, the ability to load multiple drugs, targeting specific cells or tissues, controlling drug release, and even permeation of the BBB. Hence, nanoparticle (NP) accumulation in the brain can be achieved, and the therapeutic effect can be improved. These properties make nanomedicine an attractive treatment option for brain tumors [28].

Delivery systems for nanomaterials, such as liposomes and nanostructured lipid carriers [29,30] are essential for developing new therapeutic strategies in the clinical application of TMZ [31]. NPs are a widely used modality for the delivery of chemotherapy drugs. NPs protect molecules from degradation and enhance stability and solubility, thereby improving therapeutic efficacy and specificity [32]. NPs can significantly decrease toxicity profiles significantly and elicit few side effects due to good permeability to the BBB and an enhanced permeability and retention (EPR) effect on the tumor [33].

Dual drug-loaded NPs can deliver various drugs to a tumor simultaneously, thereby obtaining a synergistic effect compared to a delivery system based on loaded with a single drug [34,35]. Combining TMZ with other drugs in dual drug-loaded NPs offers an alternative approach to developing new therapeutic strategies [36,37].

Poly (d, l-lactic-co-glycolide)-monomethoxy poly (ethylene glycol) (PLGA-mPEG) is an amphipathic polymer with hydrophilic and hydrophobic fragments. PEG and PLGA are types of biodegradable polymers approved by the US Food and Drug Administration. They are among the most commonly used biodegradable polymers for nanomedicine development due to their biocompatibility, sustained release, and low toxicity [38,39]. Additionally, mPEG can help NPs escape from phagocytosis within the reticuloendothelial system, prolong its circulation in the bloodstream, and then increase NPs accumulation in tumor tissues through the EPR effect [40,41].

We developed an innovative dual drug-loaded nanoformulation of TMZ and LIG using PLGA-mPEG for GBM treatment. The optimal proportions range of TMZ and LIG for antitumor effects were studied in vitro on the U251 cell line. A method based on the multiple-emulsion solvent evaporation was employed for the preparation of TMZ/LIG-NPs. The physicochemical properties of the NPs were determined. The viability and uptake of NPs by tumor cells in vitro were investigated to evaluate the benefit of NPs versus the use of free drugs.

## 2. Materials and Methods

### 2.1. Chemicals and Reagents

Rat C6 glioma cells and human U251 glioma cells were sourced from Zhongqiao Xinzhou Biotechnology (Shanghai, China). D-Hank’s trypsin-EDTA, dimethylsulfoxide (DMSO), poloxamer 188 (P-188), and phosphate-buffered saline (PBS) were obtained from Solarbio (Beijing, China). Egg yolk lecithin was purchased from Shanghai Advanced Vehicle Technology Pharmaceutical Ltd. (Shanghai, China, PC-98T, PC > 90%). The Cell Counting-8 (CCK-8) assay kit and tissue fixation solution were bought from Dalian Meilun Biotechnology (Dalian, China). Fetal bovine serum (FBS), F-12 K complete medium, and Dulbecco’s modified Eagle’s medium (DMEM) high-sugar complete medium were purchased from Shanghai Zhongqiao Xinzhou Biotechnology. LIG was from Chengdu Ruifensi Biotechnology (Chengdu, China). TMZ and metronidazole (MNZ) were sourced from Shanghai Yuanye Biotechnology (Shanghai, China). PLGA15K-mPEG5K was obtained from Xi’an Ruixi Biotechnology (Xi’an, China). Coumarin-6 (Cou-6) was from Aladdin Reagents (Shanghai, China). Double-distilled water was used.

### 2.2. Cell Lines and Cell Culture

U251 cells were cultured and maintained in DMEM supplemented with 10% FBS, 1% streptomycin (100 μg/mL), and 1% penicillin (100 U/mL) in a humidified atmosphere of 5% CO_2_ at 37 °C. C6 cells were cultured in F-12 K supplemented with 10% FBS in an incubator containing 5% CO_2_ at 37 °C.

### 2.3. Optimal Ratio of TMZ and LIG

The CCK-8 assay was used to assess the effects of LIG, TMZ, and their combination on the inhibition of U251 cell proliferation. DMSO was employed to dissolve LIG. Then, DMEM without FBS was employed to dilute the LIG-DMSO solution and TMZ. In this way, we prepared solutions containing a series of concentrations of LIG, TMZ, and their combination.

U251 cells were seeded in flat-bottomed 96-well microtiter plates at 2 × 10^4^/100 μL in each well. Plates were incubated at 37 °C in a humidified atmosphere containing 5% CO_2_. After 24 h, the medium was removed and replaced with a fresh medium containing increasing concentrations of each compound. After incubation for 24 h, the medium was removed, and 100 μL of fresh medium containing 10% CCK-8 was added to each well, followed by a 40 min incubation. Then, the absorbance was measured at 490 nm using a microplate reader (Multiskan™ Go; BioTek, Winooski, VT, USA). Wells containing drugs and cells were used as the sample group, and the absorbance was recorded as A_sample_. Wells with the same concentration of drugs but without seeded cells were used as a blank control, and the absorbance was recorded as A_blank_. Wells that contained cells but no sample solution were used as the negative control, and the absorbance was recorded as A_control_.

The percentage of toxic cells in the CCK-8 assay was calculated using the following equation:(1)$$\mathrm{Inhibition}\;(\%)=\frac{A_{sample}-A_{black}}{A_{control}-A_{black}}\times 100$$

The Q-value method of Jin and colleagues was used to analyze the synergistic interaction of the combination group for tumor therapy in vitro [42,43]. The formula we used was as follows:Q = E_a+b_/(E_a_ + E_b_ − E_a_ × E_b_)(2)
where E_a+b_, E_a_, and E_b_ are the effects of combination treatment, the effects of drug A alone, and the effects of drug B alone, respectively. Q < 0.85 and Q ≥ 1.15 indicate antagonism and synergy, respectively, whereas additivity is indicated by 0.85 ≤ Q < 1.15.

### 2.4. Preparation of NPs

TMZ/LIG-NPs were fabricated through a method based on multiple-emulsion solvent evaporation (Figure 2). Initially, 125 mg of PLGA 15K-PEG 5K, 5 mg of LIG, and 20 mg of TMZ were dissolved in 10 mL of dichloromethane and stirred at 500 rpm for 1 h to ensure complete dissolution. Then, the mixture was gradually added to 90 mL of a 0.15% (*w*/*v*) egg yolk lecithin solution in ethanol, stirred at 800 rpm, and sonicated in an ice bath using an ultrasonic probe (90 W) for 5 min (ultrasound: 3-s on, 3-s off) to create a primary emulsion. The former was introduced slowly into 100 mL of a 0.15% (*w*/*v*) P-188 solution with stirring at 1000 rpm to form phase 3. The phase 3 emulsion was sonicated in an ice bath with an ultrasonic probe (30 W) for 8 min (ultrasound: 3-s on, 3-s off) until homogenization to form a secondary emulsion. Next, the secondary emulsion was stirred for 24 h to evaporate the organic phase. After centrifugation (16,000 rpm, 1 h, 4 °C), TMZ/LIG-NPs were collected and washed with pure water to remove excess P-188 adhering to NP surfaces and eliminate the residual free drug. The same method was employed to produce coumarin 6-loaded nanoparticles (Cou-6-NPs).

### 2.5. Drug Loading (DL) and Encapsulation Efficiency (EE)

The TMZ/LIG-NPs were resuspended in Milli-Q^®^ water (MilliporeSigma, Burlington, MA, USA) to a volume of 25 mL and shaken well. A 2 mL aliquot of the NP suspension was transferred to a 10 mL volumetric flask, demulsified ultrasonically with methanol, passed through a 0.22 μm filter membrane, and analyzed using high-performance liquid chromatography (HPLC). EE and DL were calculated using the following equations:(3)$$DL(\%)=\frac{Amount\;of\;drug\;present\;in\;NPs}{total\;weight\;of\;NPs}\times 100$$
(4)$$EE(\%)=\frac{Amount\;of\;drug\;present\;in\;NPs}{total\;weight\;of\;the\;feeding\;drug}\times 100$$

We employed the 1260 HPLC system (Agilent Technologies, Santa Clara, CA, USA). This setup comprised a solvent-delivery system (G1311C), an autosampler (G1319B), a column oven (G1316A), and a diode array detector (G4212B). Separation was achieved on a Diamonsil^®^ C18 column (250 × 4.6 mm, 5 μm; Dikma Technologies Inc., Beijing, China). The column oven was maintained at 30 °C. The mobile phase consisted of a 0.1% solution of phosphoric acid as mobile phase A, methanol as mobile phase B, and acetonitrile as mobile phase C. A linear gradient elution program was established according to preliminary tests: 20% B and 80% A, 0–12 min; 20% C and 80% A, 12–20 min; 50–70% C, and 50–30% A, 20–35 min; 70–10% C, and 30–90% A, 35–40 min. The flow rate was maintained at 1.0 mL/min. The effluent was monitored at 280 nm. The injection volume was 10 μL.

### 2.6. Characterization

The mean particle size, polydispersity index (PDI), and zeta potential of the TMZ/LIG-NPs were determined using dynamic light scattering (DLS) employing the Zetasizer Nano ZS setup (Malvern Instruments, Malvern, UK). Measurement of each sample was performed in triplicate at 25 °C. The shape, size, and surface morphology of the TMZ/LIG-NPs were characterized by scanning electron microscopy (SEM) using the SU-4800 instrument (Hitachi, Tokyo, Japan) and transmission electron microscopy (TEM) employing the JEM1200EX system (JEOL, Tokyo, Japan). A sample (10 μL) was deposited onto carbon-supported copper meshes (300 meshes), and the excess solution was removed. The sample was completely dried and analyzed using SEM and TEM. TMZ/LIG-NPs were diluted in Milli-Q water at dilution factors of 10, 50, and 100. The mean particle size and zeta potential were measured to assess the stability of the TMZ/LIG-NPs. The NP solution was stored at 4 °C. The mean particle size and zeta potential were investigated at predetermined time intervals for 1 month. The TMZ/LIG-NPs were mixed individually with solutions of bovine serum albumin (BSA; 0.10 mg/mL), PBS containing 0.9% sodium chloride, and a solution of heparin sodium (200 U/mL). Mixtures were incubated at 37 °C for 5 h and subsequently analyzed for the respective nanotechnological parameters to assess the matrix effect of NPs.

### 2.7. In Vitro Release

In vitro release was studied in PBS (pH 7.4) using a dialysis bag. Briefly, 3.0 mL of a 50% methanol containing TMZ and LIG and a TMZ/LIG-NPs solution was sealed in a dialysis bag (molecular weight = 0.8–1.4 kDa; Solarbio) and incubated in 100 mL of release medium in stirred conditions (100 rpm) at 37 °C. We collected an aliquot (1 mL) from the release medium at predetermined time points (0.17, 0.33, 0.5, 1, 2, 4, 6, 8, 12, and 24 h). We replaced an equivalent volume of the release medium with a fresh buffer. The cumulative release of TMZ and LIG was determined using HPLC. Separation was achieved on a C18 column (250 × 4.6 mm, 5 μm; Phenomenex, Guangzhou, China). The column oven was maintained at 30 °C. The mobile phase consisted of a 0.1% solution of phosphoric acid as mobile phase A and methanol as mobile phase B. The linear gradient elution program was set according to preliminary tests: 15% B and 85% A, 0–8 min; increasing to 85% B, 8–10 min; 85% B, 10–23 min. The flow rate was maintained at 1.0 mL/min. The effluent was monitored at 267 nm for TMZ and 329 nm for LIG. The injection volume was 10 μL.

### 2.8. Plasma Pharmacokinetics and Brain Distribution

#### 2.8.1. Animals

SD rats weighing 220 ± 25 g were purchased from Speford Biotechnology Company (Beijing, China). The animal quality license is SCXK-(jing)2024–0001. All animals were maintained in accordance with the guidelines outlined by the legislation on the ethical use and care of laboratory animals.

#### 2.8.2. Grouping and Treatment

SD rats were fasted for 12 h with free access to water and were randomly divided into four groups (*n* = nine for each group, with six being dedicated to the pharmacokinetic and three for the brain distribution study). Rats were injected with TMZ/LIG-NPs, the combination of TMZ and LIG (CTL) in 4:1 and TMZ at a dose of 5 mg/kg TMZ via the tail vein. Blood samples (0.5 mL) for the pharmacokinetic study were collected from the retro-orbital plexus at time points of 0.25, 0.5, 1, 2, 3, 4, and 6 h. After centrifugation (4000 rpm, 10 min), the supernatant was collected and stored at −20 °C. Rats for brain distribution studies were sacrificed at 3 h after dosing, and brain tissue was excised. The residual blood on the rat brain tissue was carefully wiped away, and the tissue was weighed and then homogenized in an ice bath with normal saline containing 1% phosphoric acid at a ratio of 1.5 mL/g. The homogenate was centrifuged at 13,000 rpm for 10 min, and the supernatant was stored at −20 °C for later analysis.

#### 2.8.3. Blood and Tissue Sample Preparation

The 100 μL of thawed plasma sample was stabilized with 10 μL of a 10% phosphoric acid solution and added with the MNZ solution (10 μL, 10 μg/mL in acetonitrile) and 880 μL of acetonitrile. The 150 μL of thawed tissue sample was added with the MNZ solution (10 μL of 10 μg/mL in acetonitrile) and 240 μL of acetonitrile. The mixture was vortexed for 5 min and centrifuged at 13,000 rpm for 10 min. The supernatant was injected into the chromatographic system.

#### 2.8.4. Chromatographic and Mass Spectrometry Condition

The analysis was conducted using an AB liquid chromatograph–mass spectrometer (LC–MS) system using a Kinetex C18 column (100 × 3 mm, 2.6 μm; Phenomenex, CA, USA). The column oven was maintained at 40 °C; the mobile phase consisted of 0.1% formic acid in water and acetonitrile in a 92:8 ratio with a flow rate of 0.3 mL/min. The injection volume was 3 μL. The precursor ion pairs used in multiple reaction monitoring mode were 138.2–195.2 for TMZ and 82.0–172.2 for MNZ. The pharmacokinetic study was conducted and analyzed using DAS 2.0 software.

### 2.9. In Vitro Cellular Uptake

The uptake of TMZ/LIG-NPs by U251 cells was assessed using flow cytometry and fluorescence microscopy. U251 cells were seeded in 12-well dishes (1 × 10^5^ cells/well) in DMEM with 10% FBS. After incubation for 24 h, the medium was exchanged for DMEM without FBS, and cells were incubated for 12 h. Subsequently, fresh medium containing free Cou-6 and Cou-6 NPs at a Cou-6 concentration of 75 ng/mL was incubated with cells for 0.5, 1, 2, or 4 h. Cells were harvested and resuspended in PBS. Intracellular Cou-6 fluorescence was quantified using flow cytometry and analyzed with FlowJo V 10.8 (www.flowjo.com/). Cells treated for 4 h underwent three washes with cold PBS, were fixed with 4% paraformaldehyde for 20 min, and had their nuclei stained with 4′,6-diamidino-2-phenylindole solution (0.4 μg/mL) for 10 min. After three additional washes with PBS, fluorescence microscopy was performed on a fluorescence microscope (BZ-800E; Keyence, Osaka, Japan).

### 2.10. In Vitro Cytotoxicity

The cytotoxic effects of U251 and C6 cells were evaluated using the CCK-8 assay. U251 and C6 cells were seeded in flat-bottomed 96-well microtiter plates at 1 × 10^5^/100 μL in each well. The plates were incubated at 37 °C in a humidified atmosphere containing 5% CO_2_. After 24 h, the medium was aspirated, and a fresh medium containing escalating concentrations of compounds was added, maintaining a 4:1 ratio of TMZ to LIG. The CTL or TMZ/LIG-NP solution was diluted with complete medium to yield concentrations of 5, 10, 15, 20, 25, 30, 35, and 40 μg/mL for U251 cells, and 20, 30, 45, 50, 60, 70, 80, and 90 μg/mL for C6 cells. After incubation for 24 or 48 h, the medium was aspirated, and 100 μL of fresh medium containing 10% CCK-8 was added to each well, followed by incubation for 40 min. The absorbance was read at 490 nm using a microplate reader (Spark, Tecan Austria GmbH, Grodig, Austria), and the percent inhibition of samples was calculated. Nonlinear regression analysis was conducted to determine the best curve fit and the half-maximal inhibitory concentration (IC_50_).

Additionally, the matrix effect of NPs was assessed. TMZ/LIG-NPs were mixed individually with solutions of bovine serum albumin (BSA; 0.10 mg/mL), PBS containing 0.9% sodium chloride, and a solution of heparin sodium (200 U/mL). Mixtures were incubated at 37 °C for 5 h and analyzed for the respective nanotechnological parameters.

### 2.11. Statistical Analyses

Experimentally derived data are presented as the mean ± standard deviation. Statistical analyses were carried out using Prism 9.0 (GraphPad, San Diego, CA, USA) or SPSS 26.0 (IBM, Armonk, NY, USA). Data were evaluated by one-way analysis of variance, followed by the least-significant difference test to analyze differences among multiple groups compared with the control group or TMZ group. Significance was set at *p* < 0.05.

## 3. Results

### 3.1. Optimal Ratio of LIG-TMZ-Induced Toxicity Towards U251 Cells

The toxicity of LIG and TMZ against U251 glioma cells was assessed using the CCK-8 assay (Figure 3A,B). In comparison with the control group, the TMZ group demonstrated substantial inhibition of U251 cell growth within the concentration range of 50–320 µg/mL, with an IC_50_ value of 80 µg/mL. The LIG group showed growth inhibition of U251 cells within a concentration range of 5–35 µg/mL, with an IC_50_ value of 20 µg/mL.

To examine the synergistic effects between TMZ and LIG, U251 cells were treated with TMZ (40 µg/mL) and different concentrations of LIG. Q values were >1.15 (Figure 3C), indicating synergistic effects for combinations of TMZ with LIG in all groups except for LIG at 30 µg/mL. We used different concentrations of TMZ in combination with LIG, and cell-viability assays were used to examine synergistic effects. A ratio of TMZ to LIG within the range of 16:1 to 8:3 elicited synergistic effects. (Figure 3D).

### 3.2. DL and EE

The content of TMZ and LIG in TMZ/LIG-NPs was determined using HPLC after ultrasonic demulsification. The DL% of TMZ and LIG in NPs was 8.67 ± 0.87% and 2.07 ± 0.34%, and the EE% of TMZ and LIG in NPs was 52.3% and 61.5%, respectively.

### 3.3. Characterization of TMZ/LIG-NPs

The characterization, including mean particle size, PDI, zeta potential, and morphology, of NPs is closely related to their in vivo characteristics. The shape and surface morphology of TMZ/LIG-NPs are shown in Figure 4A. Most TMZ/LIG-NPs were spherical, with good dispersion and uniform particle size. The mean particle size of TMZ/LIG-NPs was 117.6 ± 0.7 nm, the PDI was 0.149 ± 0.012, and the zeta potential was −26.5 ± 0.4 mV, as measured by dynamic light scattering. TMZ/LIG-NPs exhibited high stability due to the zeta potential of NPs having a high negative value.

NP morphology was investigated using SEM (Figure 4B) and TEM (Figure 4C). The obtained mean sizes corroborated the results from dynamic light scattering. Typical TEM images showed that the NPs were spherical with internal DL. The outer layer exhibited a villous structure.

### 3.4. Stability of TMZ/LIG-NPs

We conducted tests on the stability of TMZ/LIG-NPs upon dilution and storage and the matrix effect. TMZ/LIG-NPs were diluted with Milli-Q water at 10-, 50-, and 100-fold dilutions, and the mean particle size, PDI, and zeta potential were measured. These parameters exhibited minimal changes even after a 100-fold dilution (Figure 5A), indicating the high stability of TMZ/LIG-NPs upon dilution. Furthermore, we examined the storage stability of TMZ/LIG-NPs at 4 °C over 1 month. Upon reconstitution, the stored TMZ/LIG-NPs appeared as a clear liquid without any sediment. Nanotechnological characteristics exhibited little variation during the 1-month storage (Figure 5B), illustrating high storage stability at 4 °C. This stability was attributed to the hydrophilic PEG chain, which enhanced the dispersion of nanomicelles in water and prevented their aggregation. Mixing TMZ/LIG-NPs with BSA (0.100 mg/mL), 0.9% sodium chloride in PBS solution, and heparin sodium solution (200 U/mL) individually resulted in a transparent solution without precipitation. The mean particle size and PDI (Figure 5C) showed no significant changes, suggesting that TMZ/LIG-NPs formed a stable suspension with a uniform distribution in blood. Mixing of BSA (0.100 mg/mL), 0.9% sodium chloride in PBS solution, and heparin sodium solution (200 U/mL) with NPs reduced the charge density at the NP surface, resulting in a decreased zeta potential. Nonetheless, the zeta potential remained within an acceptable range.

### 3.5. Release In Vitro

The release profiles of the CTL suspension and TMZ/LIG-NPs in PBS (pH 7.4) are presented in Figure 6. TMZ and LIG were released from the NPs in a sustained manner. In contrast with the CTL suspension, TMZ/LIG-NPs exhibited sustained release. The cumulative release of TMZ and LIG from TMZ/LIG-NPs in PBS reached 78.13% and 70.50%, respectively. In the first few hours, a typical “burst” release of TMZ from PLGA-mPEG NPs was observed, followed by a sustained release over 24 h. This effect might be attributed to the hydrophilic nature of TMZ, which ensures release by diffusion through aqueous channels from PLGA-mPEG NPs. Similar sustained release from TMZ-loaded NPs and dual drug-loaded PLGA NPs has been demonstrated in various studies [18,38]. LIG was released slowly in the phase of TMZ burst release. However, the calculation of the cumulative amounts of TMZ and LIG released at each time point revealed that the TMZ:LIG ratio was within the range of synergistic effects (Q ≥ 1.15) for release from TMZ/LIG-NPs.

### 3.6. Pharmacokinetics and Brain Distribution Study

To evaluate the in vivo behavior of TMZ/LIG-NPs accurately and their ability to penetrate the BBB, pharmacokinetics and brain distribution studies were conducted in SD rats. The mean plasma concentration–time curve of TMZ is depicted in Figure 7A, and the brain distribution is illustrated in Figure 7B, with pharmacokinetic parameters summarized in Table 1. Collectively, TMZ loading in the nanoparticles significantly improved pharmacokinetic properties and TMZ brain concentrations when compared with the TMZ group and CTL group; the t_1/2_, C_max_, AUC_0–∞_, and TMZ brain concentration were approximately 1.62-fold,1.70-fold, 4.46-fold, and 4.04-fold of those in the TMZ group, respectively. The CTL group mixture of TMZ with LIG increased the TMZ concentration in the brain, which was consistent with our previous studies. LIG can contribute to drug penetration through the BBB and provide a synergistic effect with TMZ [44]. The incorporation of drugs into NPs can facilitate their passage through the BBB and enhance drug delivery. When LIG is loaded in the NPs with TMZ together, it can significantly improve TMZ brain concentration.

### 3.7. Cellular Uptake

We investigated the potential enhancement of cellular uptake of drugs by NPs. In lieu of TMZ and LIG, Cou-6 was employed to prepare Cou-6-NPs using the methodological approach employed for TMZ/LIG-NPs. The uptake of free Cou-6 and Cou-6-NPs by U251 cells was investigated by flow cytometry (Figure 8A,B). Free Cou-6 and Cou-6-NPs exhibited a time-dependent increase in Cou-6 fluorescence. At each time point, cells treated with Cou-6-NPs displayed a significantly higher mean fluorescence intensity compared with cells treated with free Cou-6. These data indicated that NPs possessed a greater capacity for cellular uptake compared with that of free drugs. After a 4 h incubation with free Cou-6 or Cou-6-NPs, U251 cells treated with Cou-6-NPs displayed markedly stronger fluorescence in the cytoplasm than those treated with free Cou-6 (Figure 8C), indicative of enhanced cellular uptake.

### 3.8. Inhibition of U251 and C6 Cells

The CCK-8 assay was carried out to monitor the percentage inhibition in U251 and C6 cells. We treated with different concentrations of CTL or dual drug-loaded NPs for 24 or 48 h. CTL and TMZ/LIG-NPs induced concentration-dependent growth inhibition in U251 and C6 cells (Figure 9). Treatment with TMZ/LIG-NPs for 24 h reduced percentage inhibition in U251 and C6 cells significantly compared with that using CTL, with IC_50_ values of 18 μg/mL for U251 cells and 52 μg/mL for C6 cells. These values were significantly lower than the IC_50_ values for CTL in U251 cells and C6 cells (25 μg/mL and 120 μg/mL, respectively). The cytotoxic effect was more pronounced in cell cultures treated with TMZ/LIG-NPs for 48 h compared with those treated for 24 h. The cytotoxicity of NPs increased with extended co-culture times. The IC_50_ values for TMZ/LIG-NPs in U251 cells and C6 cells fell to 14 μg/mL and 44 μg/mL, respectively. The prepared TMZ/LIG-NPs exhibited higher cytotoxicity and sustained release compared to CTL.

## 4. Discussion

TMZ is widely used in GBM treatment, but it is unstable, has a short half-life, and is susceptible to degradation under physiological conditions. Furthermore, its active hydrolysis products cannot penetrate the BBB, and its indiscriminate attack on DNA in the bloodstream leads to toxic side effects, which limit its clinical application. The clinical efficacy of monotherapy relying on a single anti-tumor mechanism is limited because tumors have complex pathologies. The combined use of drugs can be of enormous value in cancer therapies [34].

Our research has shown that the combined use of LIG with TMZ for GBM treatment appears promising and warrants further investigation and development [23,25]. However, the distinct properties of LIG and TMZ present challenges for their co-delivery. Nanoformulations are believed to offer a solution to these challenges. Different nanosystems based on polymers have been used for TMZ encapsulation, among which polymeric NPs and dendrimers have been the most studied. In particular, PLGA has been the most studied polymer [45]. TMZ-PLGA NPs have been prepared by different methods to improve TMZ stability, enhance release, increase the delivery of the drug to the brain, and reduce systemic side effects [46,47,48]. PLGA can encapsulate hydrophilic agents and hydrophobic drugs [49]. PLGA-PEG can enhance dispersibility and stability in aqueous environments, which is essential for potential clinical translation [50]. NPs prepared with PLGA-PEG can form a thermodynamically stable system with improved drug accumulation through the EPR effect, leading to a prolonged presence in the circulation [41]. Encapsulating TMZ and a synergistic drug together into a dual drug-loaded nanosystem is a viable strategy for improving bioavailability and cellular uptake and ultimately enhances their anticancer activity. TMZ and curcumin co-loaded nanostructured lipid carriers have shown potential for synergistic tumor treatment while reducing side effects in vitro and in vivo [51].

We employed PEGylated PLGA (PLGA-PEG) for the development of dual drug-loaded NPs with good stability and dispersion in an aqueous solution. In terms of drug release, our results are in accordance with previously reported data [46]. NPs slowed the release of TMZ and LIG and prolonged the duration of drug action. However, they exhibited a typical initial burst release of TMZ, which was attributed to its limited solubility in water. This solubility limitation also contributed to the lower EE observed with TMZ, similar to that of other water-soluble drugs, such as insulin [52], doxorubicin [53], and cisplatin [54]. Although an initial burst release was observed in the prepared NPs, pharmacokinetic and brain distribution studies showed that encapsulation of TMZ in NPs significantly prolonged its half-life. Several pharmacokinetic parameters were markedly increased. Concurrently, the concentration of TMZ in rat brains was significantly enhanced. These observations are likely due to the augmentation of BBB permeability and the inhibition of P-gp proteins by LIG. This increases drug influx into the brain and decreases drug efflux. Meanwhile, NPs within an appropriate particle-size range enhance permeability and exert the EPR effect. These effects highlight the advantage of co-delivering TMZ and LIG, as well as the role of active ingredients from guiding meridian traditional Chinese medicine in facilitating drug entry into the brain. Cellular uptake experiments confirmed that NPs can enhance drug uptake in tumor cells. TMZ/LIG-NPs exhibited superior antitumor activity compared to free drugs and sustained anti-tumor effects in U251 and C6 glioma cell models. This increased potency can be attributed to the synergistic effect of TMZ and LIG, the enhanced cellular uptake of drugs facilitated by PLGA-PEG encapsulation, and sustained drug release, which prolongs the therapeutic duration of action and enhances the therapeutic intensity. The TMZ:LIG ratio was within the range of synergistic effects for the release from TMZ/LIG-NPs. This type of dual drug-loaded NPs, incorporating active ingredients from TCMF and chemotherapy drugs, holds promise for the treatment of malignant tumors.

## 5. Conclusions

The dual drug-loaded TMZ/LIG-NPs were prepared with PLGA-PEG, which has a uniform particle size, high compatibility with blood, and good stability during storage. NPs extend TMZ’s half-life, improve pharmacokinetics, and increase TMZ levels in the brain. Cellular uptake and cytotoxicity studies showed that TMZ/LIG-NPs increased drug uptake in tumor cells and enhanced the efficacy of anti-tumor therapy.

## Figures and Tables

**Figure 1 pharmaceutics-17-00191-f001:**
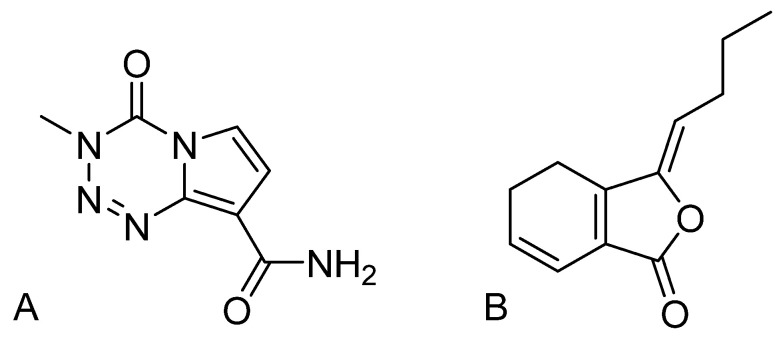
Chemical structure of (**A**) temozolomide and (**B**) ligustilide.

**Figure 2 pharmaceutics-17-00191-f002:**
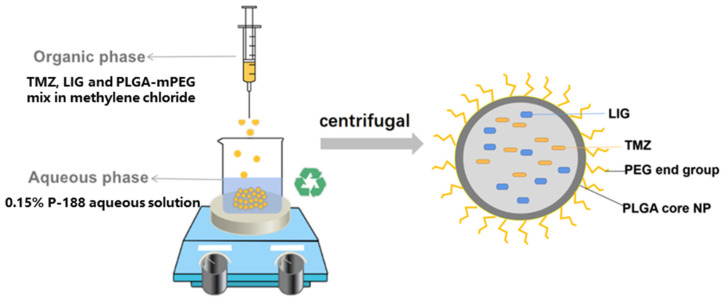
Preparation of TMZ/LIG-NPs.

**Figure 3 pharmaceutics-17-00191-f003:**
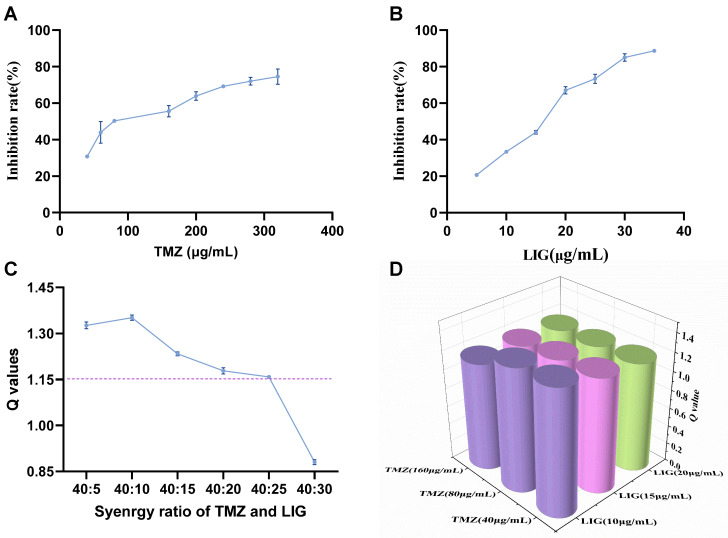
Synergistic inhibitory effects and Q values of TMZ and LIG on U251 cells. Cell inhibition of TMZ (**A**) and LIG (**B**). (**C**) Q values of TMZ (40 µg/mL) with LIG (5–30 µg/mL). (**D**) Q values of combination groups for TMZ and LIG. Values are the mean ± standard deviation (*n* = 6).

**Figure 4 pharmaceutics-17-00191-f004:**
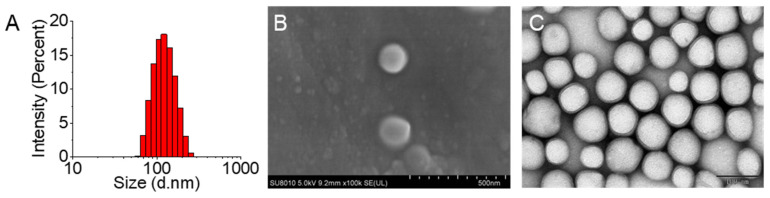
Characteristics of TMZ/LIG-NPs. (**A**) Particle-size distribution by DLS. (**B**) SEM. Scale bar = 500 nm. (**C**) TEM. Scale bar = 100 nm.

**Figure 5 pharmaceutics-17-00191-f005:**
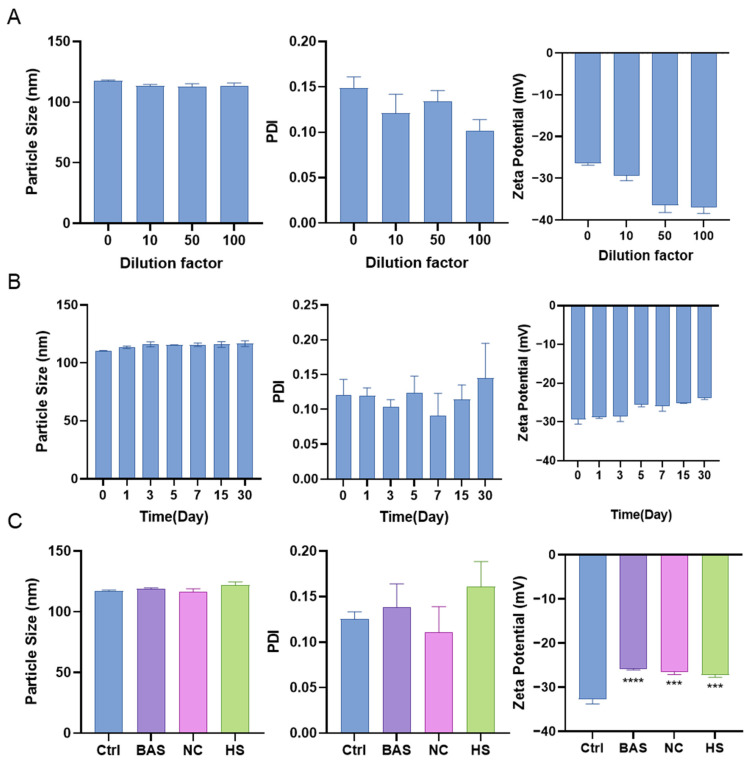
Stability upon dilution and storage and the matrix effect. (**A**) Mean particle size, PDI, and zeta potential after dilution. (**B**) Mean particle size, PDI, and zeta potential after storage (50-fold dilution). (**C**) Mean particle size, PDI, and zeta potential to test matrix effect. Values are the mean ± standard deviation (*n* = 3), *** *p* < 0.001., **** *p* < 0.0001.

**Figure 6 pharmaceutics-17-00191-f006:**
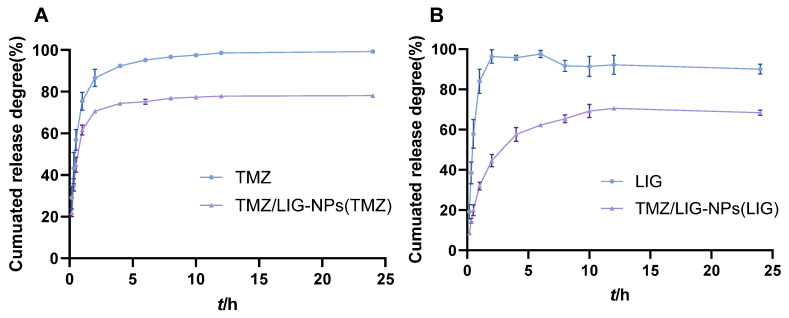
In vitro release of TMZ and LIG. (**A**) TMZ release. (**B**) LIG release. Values are the mean ± standard deviation (*n* = 3).

**Figure 7 pharmaceutics-17-00191-f007:**
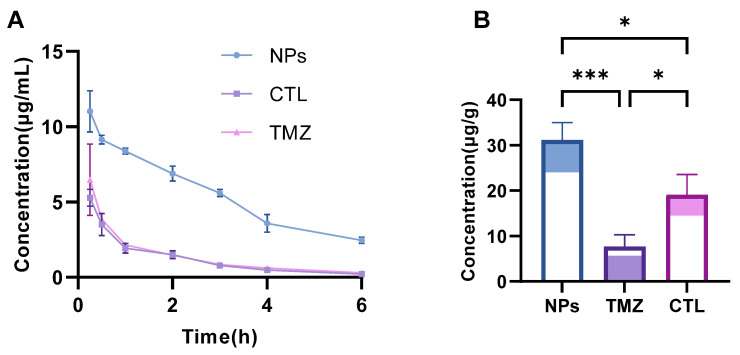
Pharmacokinetics and brain distribution. (**A**) Concentration-time profiles of TMZ in blood after injection with TMZ, CTL, TMZ/LIG-NPs via the tail vein. Values are the mean ± standard deviation (*n* = 6). (**B**) Concentration of TMZ in brain at 3 h after post-dose of TMZ, CTL, or TMZ/LIG- NPs. Values are the mean ± standard deviation (*n* = 3), * *p* < 0.05, *** *p* < 0.001.

**Figure 8 pharmaceutics-17-00191-f008:**
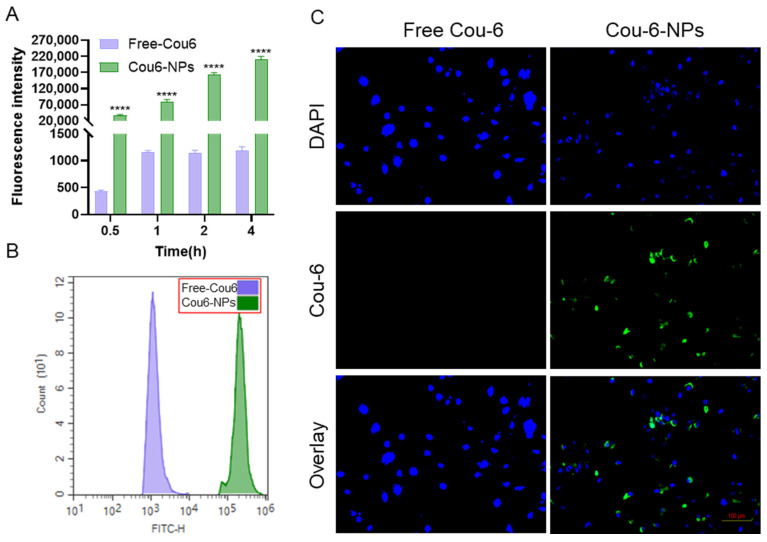
Uptake of free Cou-6 and Cou-6-NPs in U251 cells. (**A**) Time-dependent mean fluorescence intensity according to flow cytometry. (**B**) Representative FITC spectra. (**C**) Representative fluorescence-microscope images after 4-h incubation. Cell nuclei were stained with DAPI. Scale bar = 100 μm. Values are the mean ± standard deviation (*n* = 3). Differs from free Cou-6 group: **** *p* < 0.0001.

**Figure 9 pharmaceutics-17-00191-f009:**
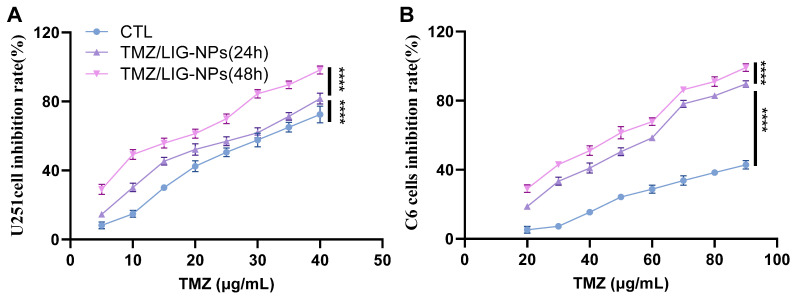
Inhibition by CTL and TMZ/LIG-NPs on U251 cells (**A**) and C6 cells (**B**). Values are the mean ± standard deviation (*n* = 3). **** *p* < 0.0001.

**Table 1 pharmaceutics-17-00191-t001:** Pharmacokinetic parameters of TMZ in blood after injection with TMZ, CTL, or TMZ/LIG-NPs via the tail vein.

	TMZ	CTL	NPs
t_1/2_ (h)	1.758 ± 0.307	1.510 ± 0.144	2.853 ± 0.384 **^##^
C_max_ (μg/mL)	6.477 ± 2.367	5.287 ± 0.554	11.027 ± 1.366 **^##^
MRT_0–∞_ (h)	1.998 ± 0.348	1.830 ± 0.164	3.969 ± 0.313 **^##^
CL (L/h/kg)	0.505 ± 0.068	0.581 ± 0.05	0.112 ± 0.008 **^##^
Vd (L/kg)	1.262 ± 0.137	1.27 ± 0.197	0.457 ± 0.036 **^##^
AUC_0–t_ (μg/mL·h)	9.263 ± 1.243	8.195 ± 0.726	34.243 ± 1.173 **^##^
AUC_0–∞_ (μg/mL·h)	10.061 ± 1.579	8.648 ± 0.702	44.953 ± 3.166 **^##^

** *p* < 0.01, versus TMZ group. ^##^
*p* < 0.01, versus CTL group.

## Data Availability

The original contributions presented in this study are included in the article. Further inquiries can be directed to the corresponding author.

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
