# Peer review of "Exploratory Study on Nanoparticle Co-Delivery of Temozolomide and Ligustilide for Enhanced Brain Tumor Therapy"

_pharmaceutics, 2025, doi:10.3390/pharmaceutics17020191_

Round 1
Reviewer 1 Report
Comments and Suggestions for Authors
The confirmation data of as prepared NPs is incomplete. Please add FT-IR and XRD graphs to confirm the synthesis.
The NPs are insoluble in water, how you investigate the release profile in PBS?
Author Response
Comments 1: The confirmation data of as prepared NPs is incomplete. Please add FT-IR and XRD graphs to confirm the synthesis.
Response 1: Thank you for your valuable feedback and suggestions regarding our manuscript. we acknowledge the importance of FT-IR and XRD in confirming the synthesis of nanoparticles. However, Ligustilide is an oily liquid and not suitable for XRD analysis. The nanoparticles we prepared were characterized for encapsulation efficiency and drug loading by organic solvent de-emulsification after centrifugation and washing, which indicates that the drug has been successfully loaded. So we didn't conduct FT-IR analysis.
Comments 2: The NPs are insoluble in water, how you investigate the release profile in PBS?
Response 2: Thank you for your valuable feedback and suggestions regarding our manuscript. we used a dialysis bag method, which is a common approach for studying the release of nano-formulations. The NPs were dispersed in a known volume of aqueous solution and placed inside a dialysis bag, then immersed in a volume of PBS that simulates the release in blood. For water-insoluble components, a certain amount of solubilizer is typically added to PBS to ensure the smooth progress of the experiment.
Reviewer 2 Report
Comments and Suggestions for Authors
The manuscript describes the approach to generating a drug delivery system and its characterisation. The topic is interesting, the protocols are written in detail, and the NPs are thoroughly described. However, a few minor points should be corrected before publication.
- Line 158. Please describe the source and purity of egg yolk lecithin. Moreover, it would be highly beneficial to provide molecular characterisation (lipid composition) of the lecithin used.
- The section 2.10 is written in a complex manner but somewhat colloquial (especially lines 268 and 279). Consider rewriting and combining the control experiment description with the main part.
- Please properly introduce the "Q value" term by clearly defining how it was calculated.
- Check all abbreviations for consistency (e.g. lines 352-353: TMZ/LIG-NPs and TMZ/LIG NPs).
- The significant part of the Discussion section sounds more like the Introduction; consider the partial moving of some sentences to the beginning of the manuscript.
- The paragraph at lines 427-435 is redundant here as it repeats some sentences from the Introduction or discusses entities unrelated to this particular work, such as quercetin.
Author Response
Comments 1: Line 158. Please describe the source and purity of egg yolk lecithin. Moreover, it would be highly beneficial to provide molecular characterisation (lipid composition) of the lecithin used.
Response 1: Thank you for pointing out the need for further detail regarding the source and purity of egg yolk lecithin used in our study. We agree with your comment and have taken the necessary steps to provide this information.Therefore, we have added the following details in the "Chemicals and Reagents" section, line 120: "Egg yolk lecithin was purchased from Shanghai Advanced Vehicle Technology Pharmaceutical Ltd (Shanghai, China, PC-98T, PC > 90%)."
Comments 2: The section 2.10 is written in a complex manner but somewhat colloquial (especially lines 268 and 279). Consider rewriting and combining the control experiment description with the main part.
Response 2: Thank you for pointing this out. We agree with your comment and have revised the section for clarity and formal scientific language. The revised section now reads as follows: "(line 273)The cytotoxic effects of U251 and C6 cells were evaluated using the CCK-8 assay. U251 and C6 cells were seeded in flat-bottomed 96-well microtiter plates at 1×105/100 μL in each well. The plates were incubated at 37°C in a humidified atmosphere containing 5% CO2. (line285)Additionally, the matrix effect of NPs was assessed. "
Comments 3: Please properly introduce the "Q value" term by clearly defining how it was calculated.
Response 3: Thank you for your comment regarding the clarification needed for the "Q value" term in our manuscript. We have taken your feedback into account and would like to provide the following explanation: The "Q value" is a method used to calculate and determine the synergistic effect of drugs, as described by the formula developed by Kim et al. We have included the formula and the criteria for assessing the synergistic effect in lines 154-159 of our manuscript. Additionally, we have referenced the relevant literature [43,43] to provide context and validation for this method. The formula we used is: Q = Ea+b/(Ea + Eb – Ea × Eb),where Ea+b, Ea, and Eb are the effect of combination treatment, effect of drug A alone, and effect of drug B alone, respectively. Q < 0.85 and Q ≥1.15 indicate antagonism and synergy, respectively, whereas additivity is indicated by 0.85≤ Q < 1.15.
Comments 4: Check all abbreviations for consistency (e.g. lines 352-353: TMZ/LIG-NPs and TMZ/LIG NPs).
Response 4: Thank you for your comment on the consistency of abbreviations within our manuscript. We have carefully reviewed all instances of abbreviations and have standardized them for consistency. All references to the nanoparticles containing TMZ and LIG are now uniformly presented as TMZ/LIG-NPs, as you suggested.
Comments 5: The significant part of the Discussion section sounds more like the Introduction; consider the partial moving of some sentences to the beginning of the manuscript.
Response 5: Thank you for your insightful comment on the structure of the Discussion section and its similarity to the Introduction. We have taken your advice and revised the manuscript to ensure that the content is appropriately distributed between the sections. Specifically, we have moved the content from line 433 of the Discussion section, which reads: "Our research has shown that LIG can enhance the anti-glioma effects of TMZ through the PI3K/Akt signaling pathway, indicating the potential of LIG to overcome resistance in GBM [23]. LIG has been shown to modulate blood-brain barrier (BBB) permeability and inhibit the P-glycoprotein (P-gp) drug-efflux system, thereby increasing drug accumulation in the brain [22]" We have integrated this information into the Introduction, now appearing as lines 73-78. This adjustment helps to clarify the purpose and significance of our study at the outset and provides a solid foundation for the subsequent discussion of our findings. We believe this change improves the flow and coherence of the manuscript.
Comments 6: The paragraph at lines 427-435 is redundant here as it repeats some sentences from the Introduction or discusses entities unrelated to this particular work, such as quercetin.
Response 6: Thank you for pointing this out. We agree with this comment. In response to your feedback, we have removed the example related to quercetin, which was not central to our study. Additionally, we have integrated the remaining content with similar points in the Introduction, specifically in lines 64-70. This revision helps to eliminate redundancy and ensures that the manuscript is concise and focused on the work presented.
We appreciate your guidance in streamlining our manuscript and maintaining the relevance of each section to the study at hand.
Best regards,
Reviewer 3 Report
Comments and Suggestions for Authors
In this manuscript, the authors characterized partially nanoparticles co-delivering Temozolomide and Ligustilide and showed that could enhance temozolomide accumulation in brain tumor, and therefore its antitumoral activity.
The study is interesting but suffer from several lacks:
- The mode of action of Ligustilide proposed when added to temozolomide should be developed more in detail in Introduction because that could help to understand synergy observed at the cellular level. As with several other natural compounds that could stick to many proteins, literature is not clear about LIG because this type of molecules appear active in many assays without consolidated argues in vivo.
- The research of optimal ratio is restricted to 1 cell line. This could be evaluated at least with a second one, potency of TMZ is highly variable from one cell line to another and it is difficult to construct all a strategy supported by data from a cell line only.
- Synergy seems to be relatively modest, and significance should be calculated.
- Ratio TMZ/LIG is specified in Fig 3 only! It is impossible to consider data of other Figures and Tables if the ratio is not clearly defined. At the top of page 379, for the realse study, it is indicated that the ratio used was within the range of synergistic effects!? What does it mean? Why the authors never indicate the ratio used at each experiment. It is clearly impossible to link data from a experiment to another, since we do not know whether the ratio is the same.
- Considering the point above, the discussion was not reviewed.
Author Response
Comments 1: The mode of action of Ligustilide proposed when added to temozolomide should be developed more in detail in Introduction because that could help to understand synergy observed at the cellular level. As with several other natural compounds that could stick to many proteins, literature is not clear about LIG because this type of molecules appear active in many assays without consolidated argues in vivo.
Response 1: Thank you for pointing this out. We have expanded the Introduction section of the manuscript, specifically at line 74, to include a description of how LIG may exert its synergistic effect against glioblastoma multiforme (GBM) by modulating the PI3K/Akt signaling pathway. Additionally, in response to the concern regarding in vivo evidence of LIG's synergistic effect, we have conducted further studies. Our results confirm that LIG can indeed reduce the required dosage of temozolomide while preserving the synergistic effect. We are in the process of preparing a manuscript that details these findings for submission for publication. We value your suggestion and remain dedicated to providing a thorough understanding of LIG's role in potentiating the efficacy of temozolomide.
Comments 2: The research of optimal ratio is restricted to 1 cell line. This could be evaluated at least with a second one, potency of TMZ is highly variable from one cell line to another and it is difficult to construct all a strategy supported by data from a cell line only.
Response 2: Thank you for pointing this out. We agree with this comment. As mentioned in our previous response, we have conducted in vivo studies that confirm the efficacy of LIG. We acknowledge the importance of validating our findings across multiple cell lines to ensure the potency of TMZ varies significantly from one cell line to another, and it is challenging to construct a comprehensive strategy based solely on data from a single cell line. In light of your feedback, we will conduct additional studies in the future that include at least one more cell line to further validate the optimal ratio of LIG and TMZ. This will provide a more robust data set and strengthen the applicability of our findings.
Comments 3: Synergy seems to be relatively modest, and significance should be calculated.
Response 3: Thank you for your comment, which addresses the synergy observed in our study and the need for a significance calculation. We have calculated the synergy using the method described by jin, which is a well-established approach in the literature. According to this method, a Q value is determined as follows: A Q value between 0.85 and 1.15 indicates an additive effect. A Q value greater than or equal to 1.15 indicates a synergistic effect. In our study, the calculated Q values exceeded 1.15, which, based on the literature method, signifies a synergistic interaction between the drugs. We have ensured that the statistical significance of these results is properly assessed and reported in the manuscript. We appreciate your attention to this detail and have made certain that our analysis and interpretation of the synergy are both rigorous and supported by the data.
Comments 4: Ratio TMZ/LIG is specified in Fig 3 only! It is impossible to consider data of other Figures and Tables if the ratio is not clearly defined. At the top of page 379, for the realse study, it is indicated that the ratio used was within the range of synergistic effects!? What does it mean? Why the authors never indicate the ratio used at each experiment. It is clearly impossible to link data from a experiment to another, since we do not know whether the ratio is the same.
Response 4: Thank you for your comment, which highlights the importance of specifying the TMZ/LIG ratio across our data presentation.
In response to your concern, we have reviewed our data and would like to clarify that according to the Q value calculated using the method by jin, a Q≥1.15 is considered to indicate a synergistic effect. As shown in Figure 3d, the ratio of TMZ to LIG within the range of 16:1 to 8:3 all resulted in Q values greater than 1.15, indicating that a synergistic effect can be achieved within this range.
For the release study, the ratio used was within the range that has been determined to have synergistic effects based on our calculations and experimental results. The NPs are designed to release both drugs in a manner that maintains their concentrations within this synergistic ratio range. Therefore, when we refer to "within the range of synergistic effects," it means that the ratio of TMZ to LIG used in the experiments is within the ratio that has been shown to elicit a synergistic response.
We apologize for any confusion caused by the initial lack of clarity regarding the ratio specification in our experiments. To ensure better understanding and data correlation, we have now explicitly stated the synergistic ratio range in line 367. We appreciate your feedback and have made the necessary revisions to improve the clarity and coherence of our manuscript.
Comments 5: Considering the point above, the discussion was not reviewed.
Response 5: Thank you for your comment. We understand that the clarity and consistency of our data presentation are crucial for a thorough review of the discussion section. We have taken the necessary steps to address the issues raised in the previous comments, particularly regarding the specification of the TMZ/LIG ratio and its impact on the interpretation of our results. We are confident that with the improved clarity of our data presentation, the discussion will now be more coherent and better support the conclusions drawn from our study. We appreciate your guidance and are committed to ensuring that our manuscript meets the highest standards of scientific rigor and clarity.
Best regards,
Round 2
Reviewer 1 Report
Comments and Suggestions for Authors
The revised version on manuscript is acceptable.
Author Response
Comments 1: The paper, while interesting, would need some further additions, in particular I believe the authors could specify the concentration of TMZ and its relationship to LIG for each experiment performed.
Response 1: Thank you for pointing this out. We understand the importance of specifying the concentrations of TMZ and its relationship to LIG for each experiment, which is crucial for the reproducibility and clarity of our research. We have addressed this by clearly indicating the specific dosages in the sections on NP preparation, cytotoxicity, and pharmacokinetic studies. Based on the results presented in section 3.1, we found that a ratio of TMZ to LIG within the range of 16:1 to 8:3 elicited synergistic effects. In our NP preparations, we maintained the ratio of TMZ and LIG within this optimal synergistic range, using TMZ as the dosing standard. For the drug combination groups, we kept the mass ratio of TMZ to LIG at 4:1. We believe these clarifications will provide a more detailed and comprehensive understanding of our experimental design and results. We have marked all the changes in red to make it easier for you to review the updates we have made.
Comments 2: I would also expect a more in-depth discussion of all the results and in particular the mechanism of BBB passage of these simple NPs.
Response 2: Thank you for pointing this out. we have made several modifications in the discussion section to provide a more comprehensive analysis of our findings. We have elaborated on the potential mechanisms by which our NPs may cross the BBB. Specifically, we discuss the role of Ligustilide (LIG) in altering BBB permeability and how the size of the prepared NPs falls within an optimal range that facilitates their passage across the BBB. All these modifications have been highlighted in red to clearly indicate the changes made in response to your feedback.
Comments 3: Minor points:
Replace TMZ on x axis of Figure 3B with LIG.
Replace synthesis of NPs with Preparation in the title of paragraph 2.4
Line 338 add a space after 4°
Response 3: Thank you for pointing out these minor points for improvement in our manuscript. We have addressed each of your suggestions, and all of these changes have been highlighted in red to clearly indicate the modifications made. We appreciate your attention to detail, and we believe these adjustments enhance the clarity and professionalism of our manuscript.